# Analysis of Intentional Electromagnetic Interference on GENEC Model Using Cylindrical Mode Matching

**DOI:** 10.3390/s23063278

**Published:** 2023-03-20

**Authors:** Wonjune Kang, No-Weon Kang, Woosang Lee, Changyul Cheon, Young-Seek Chung

**Affiliations:** 1Department of Electronics Convergence Engineering, Kwangwoon University, Seoul 01897, Republic of Korea; 2Division of Physical Metrology, Korea Research Institute of Standards and Science, Daejeon 34113, Republic of Korea; 3Agency for Defense Development, Daejeon 34186, Republic of Korea; 4School of Electrical and Computer Engineering, University of Seoul, Seoul 02504, Republic of Korea

**Keywords:** cylindrical mode matching, electromagnetic interference, electromagnetic topology, generic missile model

## Abstract

In recent times, due to the high operating frequency and low operating voltage of modern electronic devices, intended electromagnetic interference (IEMI) has been the cause of increasing damage. In particular, targets with precision electronics such as aircrafts or missiles have shown that a high-power microwave (HPM) may cause malfunction or partial destruction of the GPS or the avionic control system. Analysis of the effects of IEMI requires electromagnetic numerical analyses. However, there are limitations to conventional numerical techniques, such as the finite element method, method of moment, or finite difference time domain method, due to the complexity and large electrical length of a real target system. In this paper, we proposed a new cylindrical mode matching (CMM) technique to analyze IEMI of the generic missile (GENEC) model, which is a hollow metal cylinder with multiple apertures. Using the CMM, we can quickly analyze the effect of the IEMI inside the GENEC model from 1.7 to 2.5 GHz. The results were compared with those of the measurements and, for verification, with the FEKO, a commercial software program developed by Altair Engineering, and showed good agreement. In this paper, the electro-optic (EO) probe was used to measure the electric field inside the GENEC model.

## 1. Introduction

In recent times, due to the development of electronic technologies, electronic devices have been highly integrated and operated at a higher frequency and lower voltage. As a result, modern electronic systems have improved performances and multi-functions. However, there has been a significant increase in damage caused by malfunctions or failures resulting from intended electromagnetic interference (IEMI) [1,2,3].

Research on IEMI began with the analysis of the high-altitude electromagnetic pulse (HEMP) triggered by a nuclear explosion in the 1960s. IEMI can also be used as an electronic weapon to cause malfunctioning or permanent damage to a victim electronic system. In particular, the development of high-power microwave (HPM) sources can be a serious threat to precision electronic systems. IEMI, caused by the HPM, can lead to an increased risk of the target malfunctioning, and is especially fatal to systems that depend on a weak signal, such as GPS or radio navigation systems in aircrafts or missiles. However, since these models have a large electrical length and complex structure, it is not easy to analyze the IEMI effects using full wave analysis [4,5].

In the 1970s, Baum, Liu and Tesche proposed a generalized transmission line equation based on the electromagnetic topology (EMT), in order to quickly analyze the effect of an IEMI caused by the HPM on complex large structures such as aircrafts. It chooses the dominant penetrating routes of the IEMI source in the target and composes them as an equivalence network of a general equivalent transmission-line circuit. The equivalent network can be solved by applying the Baum-Liu-Tesche (BLT) equation [4,5,6]. The EMT analysis can combine certain analysis methods appropriate to the structure of the target and solve them rapidly and efficiently. Exploiting the equivalence network circuit and the BLT equation, the electric fields, voltages, or currents can be evaluated at the designated position. J. P. Parmantier used this method to analyze the effect of the IEMI on the cable load of the aircraft illuminated into the cockpit [7].

References [3,8] performed the EMT analyses for the susceptibility in the rectangular cavity with rectangular apertures by using mode matching and Green function. They also compared the EMT results and the experimental results in the case of multiple cavities and apertures.

Rabat analyzed the entire system based on circuit analysis to evaluate the shielding effectiveness (SE) of enclosures affected by complex apertures [9]. Similarly, Ivanov used a circuit analysis method to evaluate SE of a cylindrical connector and compared the results from the finite element method [10].

Chen introduced an equivalent circuit model to predict SE and resonant frequency of a cylindrical enclosure and introduced an EMT model to model the energy transfer inside the enclosure [11].

Cui analyzed the resonant cavity filled with various lossy dielectrics using the mode matching method. They calculated the resonance frequency and Q-factor of the cavity considering the thickness and material of the dielectric and compared the results with commercial EM tools [12]. Lima analyzed electromagnetic waves in a circular waveguide using the mode matching method. The circular waveguide was divided into each sub-domain, and the S-parameter was calculated by constructing a matrix by obtaining the boundary conditions on the coupling plane [13].

Mautz and Harrington proposed the method of moments (MoM) for the penetration of an external electromagnetic field through the aperture into a cylindrical PEC cavity [14,15,16,17,18,19]. Shumpert and Butler analyzed the internal fields inside a conducting cylinder with arbitrary contour by the penetrated field through a slot [20,21]. However, the drawback of the MoM is that if the model is of an electrically large size, a large computational time and memory may be required.

Sommerfeld and Thomas studied analytic methods for the conducting cylindrical cavity with multiple apertures, by simplifying the cylindrical PEC cavity into a two-dimensional model [22,23]. Safieddin proposed a three-dimensional analysis method when the aperture is located at the longitudinal center of the cylinder [24]. Fisahn analyzed the effects of the IEMI in the generic missile model according to the aperture size by using an experimental approach, in which the results were measured by a probe [25,26].

In [27], a whole GENEC model was divided into several sub-blocks, and S-parameters were extracted using CST MWS for each sub-block. Then, the BLT equation was applied to solve the penetrated fields.

In this paper, we proposed a cylindrical mode matching (CMM) method for the GENEC model of Diehl Co. with multiple apertures, which is a generic missile model for susceptibility analysis exposed by the IEMI. The internal circuit system and the mechanical parts were eliminated, in order to simplify the model [28]. The GENEC model has two air vent halls and two slots between the wing and body, which has a total of four apertures.

In Section 2, using the boundary conditions, we derived the formula of the total field in the aperture, and generalizes it for multiple apertures located in the arbitrary surface of the PEC cylindrical cavity. In Section 3, we compared the results of the proposed method with those of the measurements and the FEKO simulation, in order to verify the effectiveness of the proposed method.

## 2. Cylindrical Mode Matching Formulation

Figure 1 shows the GENEC model with multiple apertures, which is considered as a cylindrical PEC cavity in this paper. We assumed that the external wave is a uniform plane wave, whose polarization and angle of incidence are along the *y*-axis and the *x*-axis, respectively. The external wave is coupled through the apertures, and it affects the electromagnetic field inside the GENEC model. Then, the external wave is given as:(1)$$E^{inc}=\hat{y}E_{0}e^{jkx}\;[V/m]$$

In Figure 1, *a* indicates the radius, and 2h the longitudinal length of the GENEC model. The subscript ‘l’ refers to the *l*^th^ aperture, $2c_{l}$ the arc length of the aperture, $2d_{l}$ the z-axis length, $\Delta d_{l}$ the distance between the center of the aperture and the bottom of the GENEC model, and *L* denotes the total number of apertures on the GENEC model. In this paper, we assume that the rectangular aperture width $2c_{l}$ is sufficiently narrow to neglect the variation of field along the ϕ-axis on the aperture. Then, the external wave on the *l*^th^ aperture of the GENEC model can be expressed as follows:(2)$${\left.E\right|}_{l}^{inc}=\hat{y}E_{0}\left|\cos\phi_{l}\right|e^{jkx}e^{jka\left(1-\cos\phi_{l}\right)}\;[V/m]$$
where $E_{0}$ [V/m] means the electric field strength of the external plane wave, and $\phi_{l}$ the angle between the center of the *l*^th^ aperture and the *x*-axis.

The electric and magnetic fields inside the GENEC model can be represented as [17]:(3)$$E=E^{TE}+E^{TM}=-\nabla \times \left(\hat{z}\psi \right)+\frac{1}{j\omega \varepsilon }\nabla \times \nabla \times \left(\hat{z}\bar{\psi }\right)\;[V/m]$$
(4)$$H=H^{TE}+H^{TM}=\frac{1}{j\omega \mu }\nabla \times \nabla \times \left(\hat{z}\psi \right)+\nabla \times \left(\hat{z}\bar{\psi }\right)\;[A/m]$$

In (3) and (4), ψ and $\bar{\psi }$ denote the TE and TM potential functions, respectively, and they can be expressed as [17]:(5)$$\psi \left(\rho ,\phi ,z\right)=\sum_{m=0}^{\infty }\sum_{n=0}^{\infty }A_{mn}^{TE}J_{m}\left(\rho \Gamma_{n}\right)\cos\left(m\phi \right)\cos\left(h_{n}z\right)$$
(6)$$\bar{\psi }\left(\rho ,\phi ,z\right)=\sum_{m=0}^{\infty }\sum_{n=1}^{\infty }A_{mn}^{TM}J_{m}\left(\rho \Gamma_{n}\right)\cos\left(m\phi \right)\sin\left(h_{n}z\right)$$
where
(7)$$h_{n}\equiv \frac{n\pi }{2h}$$
(8)$$\Gamma_{n}=\left\{\begin{array}{ll} \sqrt{k^{2}-h_{n}^{2}}, & \mathrm{if}k^{2}\geq h_{n}^{2}\\ -j\sqrt{h_{n}^{2}-k^{2}}, & \mathrm{if}k^{2}<h_{n}^{2} \end{array}\right.$$

In (5) and (6), $J_{m}\left(\cdot \right)$ means the *m*^th^ order Bessel function of the first kind, and *k* the wavenumber of free space. Also, $A_{mn}^{TE}$ and $A_{mn}^{TM}$ denote the unknown coefficients for the TE and TM potential functions, respectively. If the variation of the ϕ-axis of the electromagnetic fields on the apertures can be neglected, we can also ignore the TM fields in (3) and (4). From this assumption, it is sufficient to calculate the TE fields inside the GENEC model. Therefore, in (6), $A_{mn}^{TM}\approx 0$ and $\bar{\psi }\left(\rho ,\phi ,z\right)\approx 0$.

The dominant tangential fields on the narrow rectangular aperture are electric fields with the ϕ-component $E_{\phi }^{apt}$, and the magnetic field with the z-component $H_{z}^{apt}$. Therefore, the tangential fields $E_{\phi ,l}^{apt}$ and $H_{z,l}^{apt}$ on the *l*^th^ aperture can be represented as:(9)$$E_{\phi ,l}^{apt}=\sum_{p=1}^{\infty }D_{p,l}\sin\xi_{l}\left(p,z\right)$$
(10)$$H_{z,l}^{apt}=\frac{1}{\eta_{0}}\sum_{q=1}^{\infty }C_{q,l}\sin\xi_{l}\left(q,z\right)$$
where
(11)$$\xi_{l}\left(p,z\right)\equiv \frac{p\pi }{2d_{l}}\left(z-\Delta d_{l}+d_{l}\right)$$

In (9) and (10), $D_{p,l}$ and $C_{q,l}$ denote the unknown coefficients of the *p*^th^ and *q*^th^ modes for $E_{\phi ,l}^{apt}$ and $H_{z,l}^{apt}$, respectively, on the *l*^th^ aperture. In order to calculate the unknown coefficients $D_{p,l}$, $A_{mn}^{TE}$ and $C_{q,l}$, we use the boundary conditions on the apertures. From (3), (5), and (9), the boundary condition for the electric field at ρ=a is:(12)$$\sum_{m,n}^{\infty }A_{mn}^{TE}\Gamma_{n}{J^{\prime }}_{m}\left(a\Gamma_{n}\right)\cos\left(m\phi \right)\sin\left(h_{n}z\right)=\sum_{l=1}^{L}\Omega_{l}\left(r\right)\sum_{p=1}^{\infty }D_{p,l}\sin\xi_{l}\left(p,z\right)$$
where −π≤ϕ≤π and 0≤z≤2h. Also, ${\left(\cdot \right)}^{\prime }$ denotes the derivative of the corresponding function. In (12), $\Omega_{l}\left(r\right)$ denotes the *l*^th^ aperture domain function, defined as:(13)$$\Omega_{l}\left(r\right)\equiv \left\{\begin{array}{l} 1,r\in \Omega_{l}\\ 0,\mathrm{otherwise} \end{array}\right.$$
where $\Omega_{l}$ denotes the *l*^th^ aperture region and in (12), $\sum_{m,n}^{\infty }\equiv \sum_{m=0}^{\infty }\sum_{n=1}^{\infty }$.

In order to obtain $A_{mn}^{TE}$ from (12), we define a new functional inner product as
(14)$${\left.\langle \Xi \left(\phi ,z\right),\Psi \left(\phi ,z\right)\rangle \right|}_{\left(\phi_{1},z_{1}\right)}^{\left(\phi_{2},z_{2}\right)}\equiv \int_{\phi_{1}}^{\phi_{2}}\int_{z_{1}}^{z_{2}}\Xi \left(\phi ,z\right)\Psi \left(\phi ,z\right)d\phi dz$$

We introduce the following testing function as
(15)$$\Psi_{\bar{m},\bar{n}}^{1}\left(\phi ,z\right)\equiv \cos\left(\bar{m}\phi \right)\sin\left(h_{\bar{n}}z\right)$$
and apply (12) and (15) to (14), as
(16)$${\left.\langle (12),\Psi_{\bar{m},\bar{n}}^{1}\left(\phi ,z\right)\rangle \right|}_{\left(-\pi ,0\right)}^{\left(\pi ,2h\right)}$$

Equation (16), in detail, is as follows
(17)$$\begin{array}{cc} \sum_{m,n}^{\infty } & \left[\begin{array}{l} A_{mn}^{TE}\Gamma_{n}{J^{\prime }}_{m}\left(a\Gamma_{n}\right)\int_{-\pi }^{\pi }\cos\left(m\phi \right)\cos\left(\bar{m}\phi \right)d\phi \\ \;\times \int_{0}^{2h}\sin\left(h_{n}z\right)\sin\left(h_{\bar{n}}z\right)dz \end{array}\right]\\ & =\sum_{l=1}^{L}\sum_{p=1}^{\infty }\left[\begin{array}{l} D_{p,l}\int_{-\zeta_{l}}^{\zeta_{l}}\cos\left(\bar{m}\phi \right)d\phi \\ \;\times \int_{\Delta d_{l}-d_{l}}^{\Delta d_{l}+d_{l}}\sin\left(h_{\bar{n}}z\right)\sin\xi_{l}\left(p,z\right)dz \end{array}\right] \end{array}$$
where $\zeta_{l}\equiv c_{l}/a$ means the radian half angle of the *l*^th^ aperture. Rearranging (17) using the orthogonality of the trigonometric function, we get
(18)$$A_{\bar{m}\bar{n}}^{TE}\Gamma_{\bar{n}}{J^{\prime }}_{\bar{m}}\left(a\Gamma_{\bar{n}}\right)=\frac{1}{\pi h}\sum_{l=1}^{L}\sum_{p=1}^{\infty }D_{p,l}U_{1,l}\left(\bar{m}\right)U_{2,l}\left(\bar{n},p\right)$$
where
(19)$$U_{1,l}\left(\bar{m}\right)\equiv \int_{-\zeta_{l}}^{\zeta_{l}}\cos\left(\bar{m}\phi \right)d\phi$$
(20)$$U_{2,l}\left(\bar{n},p\right)\equiv \int_{\Delta d_{l}-d_{l}}^{\Delta d_{l}+d_{l}}\sin\left(h_{\bar{n}}z\right)\sin\xi_{l}\left(p,z\right)dz$$

Similarly, using (4), (5), and (10), we can represent the tangential magnetic field on the *l*^th^ aperture $H_{z,l}^{apt}$ as
(21)$$\frac{1}{jk}\sum_{m,n}^{\infty }A_{mn}^{TE}\Gamma_{n}^{2}J_{m}\left(a\Gamma_{n}\right)\cos\left(m\phi \right)\sin\left(h_{n}z\right)=\sum_{q=1}^{\infty }C_{q,l}\sin\xi_{l}\left(q,z\right)$$

In order to obtain $C_{q,l}$ from (21), we use the following testing function
(22)$$\Psi_{\bar{q}}^{2}\left(\phi ,z\right)\equiv \sin\xi_{l}\left(\bar{q},z\right)$$
and apply the functional inner product as
(23)$${\left.\langle (21),\Psi_{\bar{q}}^{2}\left(\phi ,z\right)\rangle \right|}_{\left(-\zeta_{l},\Delta d_{l}-d_{l}\right)}^{\left(\zeta_{l},\Delta d_{l}+d_{l}\right)}$$

Equation (23) can be rewritten in detail as follows
(24)$$\begin{array}{cc} \frac{1}{jk} & \sum_{m,n}^{\infty }\left[\begin{array}{l} A_{mn}^{TE}\Gamma_{n}^{2}J_{m}\left(a\Gamma_{n}\right)\int_{-\zeta_{l}}^{\zeta_{l}}\cos\left(m\phi \right)d\phi \\ \;\times \int_{0}^{2h}\sin\left(h_{n}z\right)\sin\xi_{l}\left(\bar{q},z\right)dz \end{array}\right]\\ & =\sum_{q=1}^{\infty }\left[\begin{array}{l} C_{q,l}\int_{-\zeta_{l}}^{\zeta_{l}}d\phi \\ \times \int_{\Delta d_{l}-d_{l}}^{\Delta d_{l}+d_{l}}\sin\xi_{l}\left(q,z\right)\sin\xi_{l}\left(\bar{q},z\right)dz \end{array}\right] \end{array}$$

Rearranging (24) in terms of $C_{\bar{q},l}$, we get
(25)$$\frac{a}{2\zeta_{l}d_{l}}\frac{1}{jk}\sum_{m,n}^{\infty }\left[\begin{array}{l} A_{mn}^{TE}\Gamma_{n}^{2}J_{m}\left(a\Gamma_{n}\right)\\ \;\times U_{1,l}\left(m\right)U_{2,l}\left(n,\bar{q}\right) \end{array}\right]=C_{\bar{q},l}$$

The cylindrical transform of the tangential electric field in the GENEC model is given as [17]
(26)$${\bar{E}}_{\phi }\left(r,\alpha \right)=\frac{1}{2\pi }\int_{-\infty }^{\infty }\int_{0}^{2\pi }E_{\phi }\left(a,\phi ,z\right)e^{-jr\phi }e^{-j\alpha z}d\phi dz$$
where
(27)$$E_{\phi }\left(a,\phi ,z\right)=\Omega_{l}\left(r\right)\sum_{p=1}^{\infty }D_{p,l}\sin\xi_{l}\left(p,z\right)$$

By substituting (27) into (26), we get
(28)$$\begin{array}{cc} {\bar{E}}_{\phi }\left(r,\alpha \right) & =\frac{1}{2\pi }\sum_{p=1}^{\infty }\left[\begin{array}{l} D_{p,l}\int_{-\zeta_{l}/a}^{\zeta_{l}/a}e^{-jr\phi }d\phi \\ \;\times \int_{\Delta d_{l}-d_{l}}^{\Delta d_{l}+d_{l}}e^{-j\alpha z}\sin\xi_{l}\left(p,z\right)dz \end{array}\right]\\ & =\frac{1}{2\pi }\sum_{p=1}^{\infty }D_{p,l}U_{3}\left(r\right)U_{4}\left(\alpha ,p\right) \end{array}$$
where
(29)$$U_{3,l}\left(r\right)\equiv \int_{-\zeta_{l}}^{\zeta_{l}}e^{-jr\phi }d\phi$$
(30)$$U_{4,l}\left(\alpha ,p\right)\equiv \int_{\Delta d_{l}-d_{l}}^{\Delta d_{l}+d_{l}}e^{-j\alpha z}\sin\xi_{l}\left(p,z\right)dz$$

Also, the inverse transform of (26) is shown in (31)
(31)$$\begin{array}{cc} E_{\phi }\left(a,\phi ,z\right) & =\frac{1}{2\pi }\sum_{r=-\infty }^{\infty }e^{jr\phi }\int_{-\infty }^{\infty }{\bar{E}}_{\phi }\left(r,\alpha \right)e^{j\alpha z}d\alpha \\ & =\frac{1}{2\pi }\sum_{r=-\infty }^{\infty }e^{jr\phi }\int_{-\infty }^{\infty }\left[\begin{array}{l} g_{r}\left(\alpha \right)\sqrt{k^{2}-\alpha^{2}}\\ \;\times {H_{r}^{\left(2\right)}}^{\prime }\left(a\sqrt{k^{2}-\alpha^{2}}\right)e^{j\alpha z} \end{array}\right]d\alpha \end{array}$$
where ${H_{r}^{\left(2\right)}}^{\prime }\left(\cdot \right)$ is the derivative of the 2nd kind Hankel function, and $g_{r}\left(\alpha \right)$ is a function to be determined from the boundary conditions [17].

By substituting (28) into (31), and rearranging it, we then get
(32)$$\frac{1}{2\pi }\sum_{p=1}^{\infty }D_{p,l}U_{3,l}\left(r\right)U_{4,l}\left(\alpha ,p\right)=\beta g_{r}\left(\alpha \right){H_{r}^{\left(2\right)}}^{\prime }\left(a\beta \right)$$
where
(33)$$\beta =\left\{\begin{array}{ll} \;\sqrt{k^{2}-\alpha^{2}}, & \mathrm{if}k^{2}-\alpha^{2}\geq 0\\ -j\sqrt{\alpha^{2}-k^{2}}, & \mathrm{if}k^{2}-\alpha^{2}<0 \end{array}\right.$$

Finally, the total magnetic field on the *l*^th^ aperture $H_{z,l}^{apt}$ can be represented as
(34)$$\begin{array}{cc} e^{-jka\cos\left(\phi \right)} & -\sum_{w=-\infty }^{\infty }j^{-w}{J^{\prime }}_{w}\left(ka\right)\Lambda_{w}\left(ka\right)e^{jw\phi }\\ & +\frac{1}{jk}\sum_{r=-\infty }^{\infty }e^{jr\phi }\int_{-\infty }^{\infty }\beta^{2}e^{j\alpha z}g_{r}\left(\alpha \right)H_{r}^{\left(2\right)}\left(a\beta \right)d\alpha \\ & =\sum_{q=1}^{\infty }C_{q,l}\sin\xi_{l}\left(q,z\right) \end{array}$$
(35)$$\Lambda_{w}\left(x\right)\equiv \frac{H_{w}^{\left(2\right)}\left(x\right)}{{H_{w}^{\left(2\right)}}^{\prime }\left(x\right)}$$
where $H_{r}^{\left(2\right)}\left(\cdot \right)$ is the 2nd kind Hankel function. Rearranging (34) in terms of $C_{q,l}$, we use the following testing function
(36)$$\Psi_{\bar{q}}^{3}\left(\phi ,z\right)\equiv \sin\xi_{l}\left(\bar{q},z\right)$$
and apply the functional inner product to (34) as
(37)$${\left.\langle (34),\Psi_{\bar{q}}^{3}\left(\phi ,z\right)\rangle \right|}_{\phi =-\zeta_{l},z=\Delta d_{l}-d_{l}}^{\phi =\zeta_{l},z=\Delta d_{l}+d_{l}}$$

Similarly, using (36) and (37), we get
(38)$$\begin{array}{cc} \left[\begin{array}{l} \int_{-\zeta_{l}}^{\zeta_{l}}e^{-jka\cos\phi }d\phi \\ -\sum_{w=-\infty }^{\infty }j^{-w}{J^{\prime }}_{w}\left(ka\right)\Lambda_{w}\left(ka\right)U_{5,l}\left(w\right) \end{array}\right]\frac{aU_{6,l}\left(\bar{q}\right)}{2\zeta_{l}d_{l}} & \\ \;+\frac{a}{jk2\zeta_{l}d_{l}}\sum_{r=-\infty }^{\infty }\left\{\begin{array}{l} U_{5,l}\left(r\right)\\ \times \int_{-\infty }^{\infty }\left[\begin{array}{l} \beta^{2}g_{r}\left(\alpha \right)U_{7,l}\left(\alpha ,\bar{q}\right)\\ \;\times H_{r}^{\left(2\right)}\left(a\beta \right) \end{array}\right]d\alpha \end{array}\right\} & =C_{\bar{q},l}^{} \end{array}$$
where
(39)$$U_{5,l}\left(r\right)\equiv \int_{-\zeta_{l}}^{\zeta_{l}}e^{jr\phi }d\phi$$
(40)$$U_{6,l}\left(\bar{q}\right)\equiv \int_{\Delta d_{l}-d_{l}}^{\Delta d_{l}+d_{l}}\sin\xi_{l}\left(\bar{q},z\right)dz$$
(41)$$U_{7,l}\left(\alpha ,\bar{q}\right)\equiv \int_{\Delta d_{l}-d_{l}}^{\Delta d_{l}+d_{l}}e^{j\alpha z}\sin\xi_{l}\left(\bar{q},z\right)dz$$

Equations (18), (25), (32), and (38) can be expanded from (42) to (45), by considering all *L* apertures as
(42)$$A_{\bar{m}\bar{n}}^{TE}\Gamma_{\bar{n}}{J^{\prime }}_{\bar{m}}\left(a\Gamma_{\bar{n}}\right)=\frac{1}{\pi h}\sum_{l=1}^{L}\sum_{p=1}^{\infty }D_{p,l}^{}U_{1,l}^{}\left(\bar{m}\right)U_{2,l}^{}\left(\bar{n},p\right)$$
(43)$$\frac{a}{2\zeta_{\bar{l}}d_{\bar{l}}}\frac{1}{jk}\sum_{m,n}^{\infty }\left[\begin{array}{l} A_{mn}^{TE}\Gamma_{n}^{2}J_{m}\left(a\Gamma_{n}\right)\\ \;\times U_{1,\bar{l}}^{}\left(m\right)U_{2,\bar{l}}^{}\left(n,\bar{q}\right) \end{array}\right]=C_{\bar{q},\bar{l}}^{}$$
(44)$$\frac{1}{2\pi }\sum_{l=1}^{L}\sum_{p=1}^{\infty }D_{p,l}^{}U_{3,l}^{}\left(r\right)U_{4,l}^{}\left(\alpha ,p\right)=\beta g_{r}\left(\alpha \right)H_{r}^{\left(2\right)}{}^{\prime }\left(a\beta \right)$$
(45)$$\begin{array}{cc} \left[\begin{array}{l} \int_{-\zeta_{\bar{l}}}^{\zeta_{\bar{l}}}e^{-jka\cos\left(\phi \right)}d\phi \\ -\sum_{w=-\infty }^{\infty }j^{-w}{J^{\prime }}_{w}\left(ka\right)\Lambda_{w}\left(ka\right)U_{5,\bar{l}}^{}\left(w\right) \end{array}\right]\frac{aU_{6,\bar{l}}^{}\left(\bar{q}\right)}{2\zeta_{\bar{l}}d_{\bar{l}}} & \\ \;+\frac{a}{jk2\zeta_{\bar{l}}d_{\bar{l}}}\sum_{r=-\infty }^{\infty }U_{5,\bar{l}}^{}\left(r\right)\int_{-\infty }^{\infty }\left[\begin{array}{l} \beta^{2}g_{r}\left(\alpha \right)U_{7,\bar{l}}^{}\left(\alpha ,\bar{q}\right)\\ \;\times H_{r}^{\left(2\right)}\left(a\beta \right) \end{array}\right]d\alpha & =C_{\bar{q},\bar{l}}^{} \end{array}$$
where $\bar{l}=1,2,3,\ldots ,L$.

Considering the finite number of modes, (42)–(45) can be reorganized into the following matrix form as
(46)[Λ][D]=[V]
where
(47)$$\left[\Lambda \right]=\left[\begin{array}{cccc} \left\{\lambda_{11}\right\} & \left\{\lambda_{12}\right\} & \ldots & \left\{\lambda_{1L}\right\}\\ \left\{\lambda_{21}\right\} & \left\{\lambda_{22}\right\} & \ldots & \vdots \\ \vdots & \vdots & \ddots & \vdots \\ \left\{\lambda_{L1}\right\} & \ldots & \ldots & \left\{\lambda_{LL}\right\} \end{array}\right]$$
(48)$$\left[D\right]=\left[\begin{array}{c} \left\{D_{1}\right\}\\ \left\{D_{2}\right\}\\ \vdots \\ \left\{D_{L}\right\} \end{array}\right]$$
(49)$$[V]=\left[\begin{array}{c} \left\{V_{1}\right\}\\ \left\{V_{2}\right\}\\ \vdots \\ \left\{V_{L}\right\} \end{array}\right]$$
(50)$$\begin{array}{cc} {\left\{\lambda_{\bar{l}l}^{}\right\}}_{\bar{q},p}= & \frac{1}{jk\pi h}\sum_{m,n}^{M,N}\left[\begin{array}{l} U_{1,l}^{}\left(m\right)U_{2,l}^{}\left(n,p\right)\\ \times U_{1,\bar{l}}^{}\left(m\right)U_{2,\bar{l}}^{}\left(n,\bar{q}\right)\Gamma_{n}^{}\frac{J_{m}\left(a\Gamma_{n}\right)}{{J^{\prime }}_{m}\left(a\Gamma_{n}\right)} \end{array}\right]\\ & -\frac{1}{jk\pi }\sum_{r=-R}^{R}U_{3,l}^{}\left(r\right)U_{5,\bar{l}}^{}\left(r\right)I\left(r,p,\bar{q},l,\bar{l}\right) \end{array}$$
(51)$${\left\{V_{\bar{l}}\right\}}_{\bar{q}}=\left[\begin{array}{l} \int_{-\zeta_{\bar{l}}}^{\zeta_{\bar{l}}}e^{-jka\cos\left(\phi \right)}d\phi \\ \;-\sum_{w=-W}^{W}j^{-w}{J^{\prime }}_{w}\left(ka\right)\Lambda_{w}\left(ka\right)U_{5,\bar{l}}^{}\left(w\right) \end{array}\right]U_{6,\bar{l}}^{}\left(\bar{q}\right)$$
(52)$$I\left(r,p,\bar{q},l,\bar{l}\right)\equiv \int_{0}^{\infty }U_{4,l}^{}\left(\alpha ,p\right)U_{7,\bar{l}}^{}\left(\alpha ,\bar{q}\right)\Lambda_{r}\left(a\beta \right)\beta d\alpha$$

In (47)–(49), $\left\{\lambda_{\bar{l}l}^{}\right\}$ and $\left\{V_{\bar{l}}^{}\right\}$ are the system and forcing block matrices with $Q_{\bar{l}}\times P_{l}$ and $Q_{\bar{l}}\times 1$, respectively. Also, $\left\{D_{l}^{}\right\}$ is the unknown block matrix with $P_{l}\times 1$. Here, $P_{l}$ and $Q_{\bar{l}}$ denote the finite number of modes for $E_{\phi ,l}^{apt}$ of the $l^{th}$ aperture, and $H_{z,\bar{l}}^{apt}$ of the ${\bar{l}}^{th}$ aperture, respectively. In this paper, we determined the minimum number of modes for the $l^{th}$ aperture as follows
(53)$$P_{l}\geq 5\frac{d_{l}}{\lambda_{\min}}+1$$
where $d_{l}$ denotes the length of the *l*^th^ aperture, and $\lambda_{\min}$ the lowest wavelength in free space of the designated frequency band. Also, for convenience of calculation, we use $P_{l}=Q_{\bar{l}}$ when $l=\bar{l}$. In (50) and (51), the summation indices *R* and *W* are chosen as 30.

By solving (46), we can obtain $\left\{D_{l}\right\}$ at the *l*^th^ aperture and obtain $A_{{}_{mn}}^{TE}$ from (12). Finally, the electromagnetic fields inside the GENEC model can be calculated using (3) and (4).

In (50), the calculation of ${\left\{\lambda_{\bar{l}l}\right\}}_{\bar{q},p}$ requires the infinite integration of (52). To handle the infinite integration, we test whether $\Lambda_{r}\left(x\right)$ has poles or singularities. Using the recursive relationship, when x→0, $\Lambda_{r}\left(x\right)$ converges as follows [29,30]
(54)$$\underset{x\to 0}{\lim}\Lambda_{r}\left(x\right)=\underset{x\to 0}{\lim}\frac{1}{\frac{-r}{x}+\frac{H_{r-1}^{\left(2\right)}\left(x\right)}{H_{r}^{\left(2\right)}\left(x\right)}}=0$$

Also, when x>>1, $H_{r}^{\left(2\right)}\left(x\right)$ and ${H_{r}^{\left(2\right)}}^{\prime }\left(x\right)$ can be represented, respectively, as
(55)$$H_{r}^{\left(2\right)}\left(x\right)\sim \sqrt{\frac{2}{\pi x}}e^{-j\left[x-\left(r+\frac{1}{2}\right)\frac{\pi }{2}\right]}$$
(56)$${H_{r}^{\left(2\right)}}^{\prime }\left(x\right)∼\frac{1}{2}\sqrt{\frac{2}{\pi x}}\left[e^{-j\left[x-\left(r-\frac{1}{2}\right)\frac{\pi }{2}\right]}-e^{-j\left[x-\left(r+\frac{3}{2}\right)\frac{\pi }{2}\right]}\right]$$

From (55) and (56), when x→∞, $\Lambda_{r}\left(x\right)$ converges as follows
(57)$$\underset{x\to \infty }{\lim}\Lambda_{r}\left(x\right)=j$$

Therefore, the integrand of (52) has no singularity. The integrand of (52) can be rewritten as
(58)$$G\left(\alpha ,r,p,\bar{q},l,\bar{l}\right)=U_{4,l}^{}\left(\alpha ,p\right)U_{7,\bar{l}}^{}\left(\alpha ,\bar{q}\right)\Lambda_{r}\left(a\beta \right)\beta$$

Equation (58) has zeros when $\alpha_{0}$ is equal to pπ/d, $\left(p+0.5\right)\pi /d$, $\bar{q}\pi /d$ or $\left(\bar{q}+0.5\right)\pi /d$. Here, the set of $\alpha_{0}$ to satisfy $G\left(\alpha_{0},r,p,\bar{q},l,\bar{l}\right)=0$ can be represented as
(59)$$\left\{\alpha_{0}\right\}=\left\{\alpha_{0,0},\alpha_{0,1},\ldots ,\alpha_{0,j},\ldots \right\}$$
where $\alpha_{0,j}$ is the *j*^th^ zero of (58), and is sorted to become $\alpha_{0,j}<\alpha_{0,k}$ for j<k.

Using the adjacent two zeros of the integrand of (52), the equation can be rewritten as
(60)$$I\left(r,p,\bar{q},l,\bar{l}\right)=\sum_{j=0}^{\infty }\int_{\alpha_{0,j}}^{\alpha_{0,j+1}}G\left(\alpha ,r,p,\bar{q},l,\bar{l}\right)d\alpha$$

Since $G\left(\alpha ,r,p,\bar{q},l,\bar{l}\right)$ has no pole, we can apply the Gaussian quadrature integration to (60).

## 3. Simulation and Measurement Results

Figure 2 shows the simplified GENEC model, which is made with the GENEC (Generic missile) system of German company, Diehl [28].

The GENEC model measures 657.5 mm in length and 98.5 mm in diameter and has 4 apertures. Table 1 shows the sizes and locations of the apertures. In an ideal case, the first resonance frequency is approximately 1.79 GHz. During the simulation in the proposed method, the apertures were considered to be rectangles with the same areas.

Figure 3 shows that in order to measure the effect of the IEMI inside the GENEC model, we construct the measurement system in an anechoic chamber. The frequency range is set as 1.7~2.5 GHz, considering the first resonance frequency of the model. For the radiating antenna, we use a double-ridged horn antenna made by ETS LINDGREN Model 3117, whose frequency ranges from 1 to 18 GHz, which has the VSWR of 2:1 in 1.7~2.5 GHz. The radiating antenna has linear polarization in the *y*-axis, and Figure 3 shows the incident angle is perpendicular to the GENEC model. The distance between the radiating antenna and the center of the GENEC model is 1.5 m.

In order to measure the internal electric field strength inside the GENEC model, we use an electro-optic (EO) probe. The EO probe is a photonic-assisted electric field probe that has been widely used in various electromagnetic sensing applications. EO probes have fairly low invasiveness, high-power robustness, and wideband nature of EO crystals. EO crystals are associated with ultrafast sampling laser pulses and can be used to realize the near-field characterization of antennae [31], and also high-power microwaves (HPM) [32]. EO probes can handle over an MV/m scale field [33] and offer a 100 dB dynamic range with excellent linearity [34,35].

Figure 4 shows that it was easier to install inside the GENEC model. Also, a 0.5 mm × 1.5 mm × 1.5 mm LiTaO_3_ piece was used as an EO crystal. The EO crystal works as an E-field sensor along a single direction on the optic axis.

The optical fiber was connected to EO measurement system which is located outside of the chamber. The system consisted of an optical circulator, a polarization controller, a DFB (Distributed Feedback) laser with a TEC (Thermoelectric Cooler) controller, an optical power meter and a photo detector with an amplifier. The output of a system was connected to the spectrum analyzer.

In this paper, the optic axis was placed along the y-axis inside the GENEC model. The optic axis of the sensor can be changed accordingly, in order to measure fields in other directions. Figure 5 shows the measurement system, including the EO probe [28]. The E-field strength from an EO probe at a distance of 1.5 m from the antenna was measured, in order to calibrate the sensor’s EO signal strength. The shielding effectiveness inside the GENEC model was measured using a calibrated EO probe, and this allowed the field strength to be obtained.

The results of the measurement and the simulation are represented by the shielding effectiveness (SE), as follows [27,36]
(61)$$\mathrm{SE}\equiv 20\log\left(\frac{\left|E_{inc}\right|}{\left|E_{cav}\right|}\right)\;[\mathrm{dB}]$$

In (61), $\left|E_{inc}\right|$ and $\left|E_{cav}\right|$ represent the strength of the external electric field, and the electric field inside the GENEC model, respectively. This equation can be used to obtain the shielding efficiency for the target of each frequency and predict the frequency band in which the target is weakened.

Figure 6 shows the two EO probes that are located at P1 and P2 inside the GENEC model. They are located 181.25 and 79.85 mm away, respectively, from the center of the GENEC model. Figure 7 shows our comparison of the results of the proposed method, the experiment, and the simulation using the FEKO. The range of frequency was 1.7~2.5 GHz, and 801 frequency samples were used in the simulation.

From 1.7 to 2.5 GHz, the results of the proposed method show good agreement with those of the measurements and the commercial EM tool. Also, Table 2 shows that the computing time of the proposed method is up to 100 times faster than the commercial EM tool. The CPU was an Intel Core i7-2600k @3.4 GHz, and 16 GB RAM was used for the two simulations.

## 4. Conclusions

In this paper, we proposed an intentional electromagnetic interference (IEMI) analysis for a cylindrical PEC cavity with multiple apertures based on cylindrical mode matching (CMM) and the electromagnetic topology (EMT) when the target is exposed by an external electromagnetic wave. When the model is electrically large, it is very difficult to apply full wave analysis methods, such as FEM, FDTD, or MoM, to the IEMI analysis. To overcome this problem, Baum proposed the EMT, which is an approximate general transmission line method. Since the previous studies for the IEMI effect on PEC structure based on the EMT have related to the rectangular PEC cavity with multiple apertures, we extended it to the IEMI analysis for the cylindrical PEC cavity, which is a generic missile model.

For the measurement of electric fields inside the GENEC model, we used the electro-optic probe, which can handle over an MV/m scale field and operate a wide dynamic range with excellent linearity.

The results of the proposed method were compared with those of the measurements and the full wave analysis and showed good agreement. In addition, the proposed method was shown to be about 100 times faster than the FEKO.

## Figures and Tables

**Figure 1 sensors-23-03278-f001:**
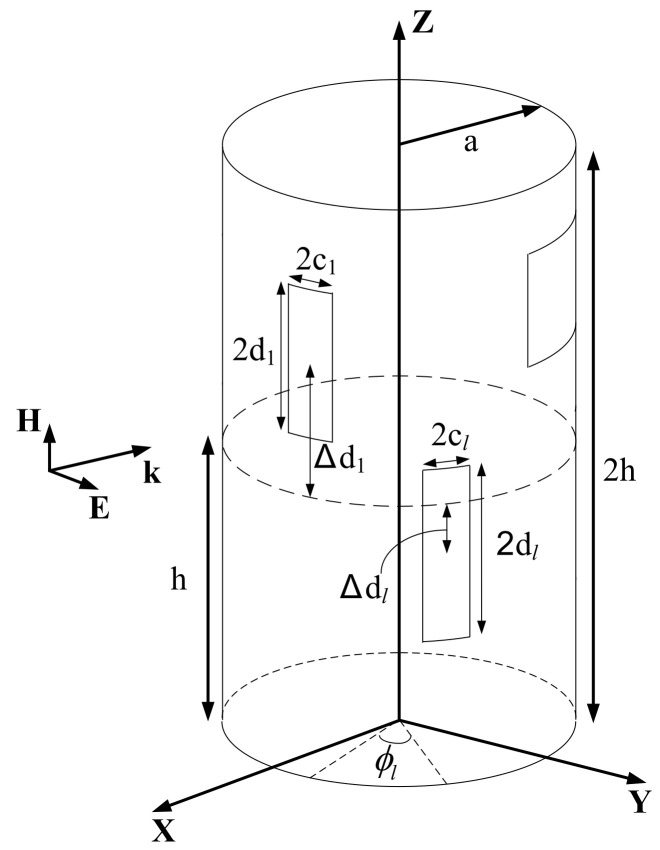
The geometry of the GENEC model with multiple apertures.

**Figure 2 sensors-23-03278-f002:**
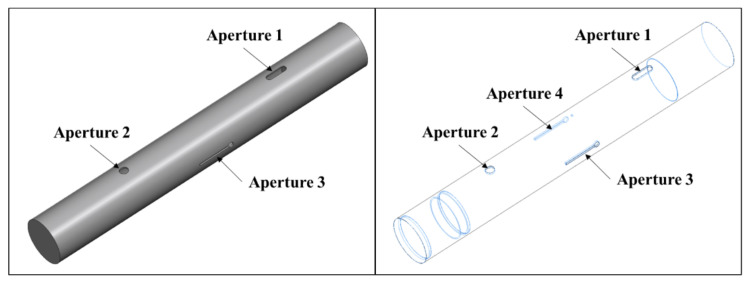
Simplified GENEC model with 4 apertures.

**Figure 3 sensors-23-03278-f003:**
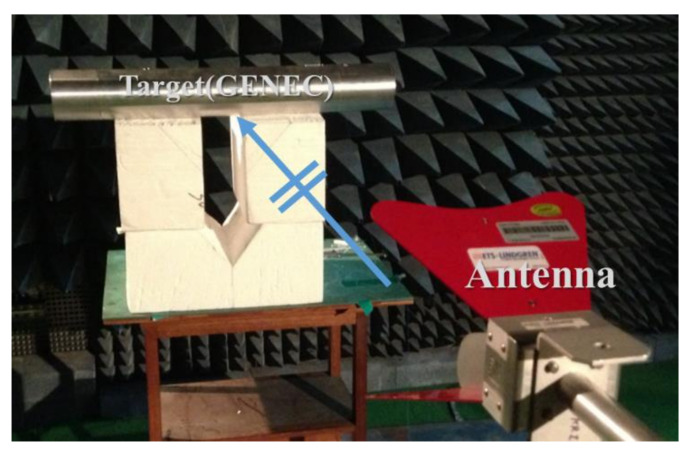
Experiment for the Simplified GENEC model by an external incident wave.

**Figure 4 sensors-23-03278-f004:**
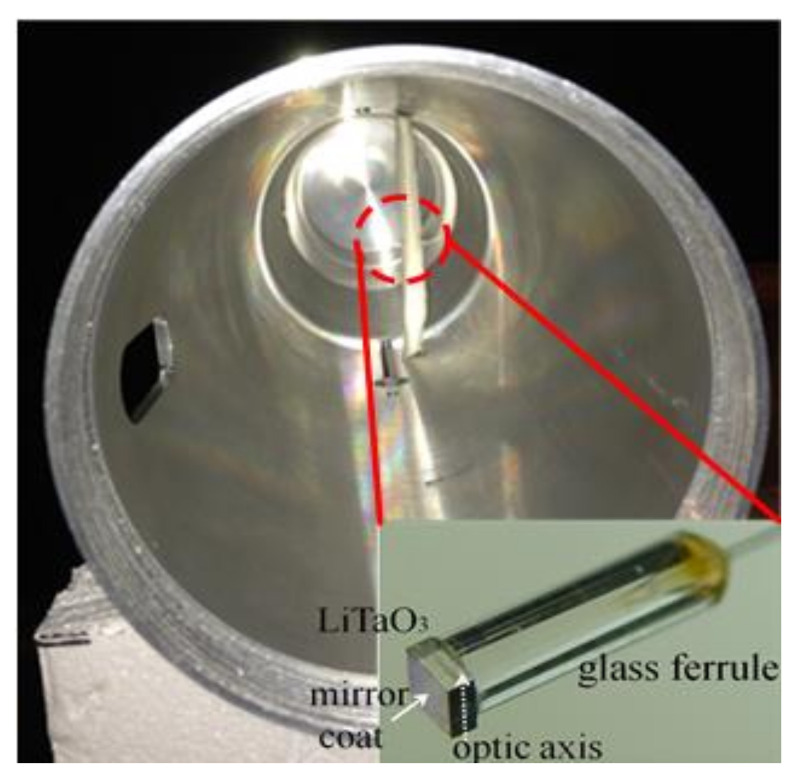
EO sensor probe in the GENEC model.

**Figure 5 sensors-23-03278-f005:**
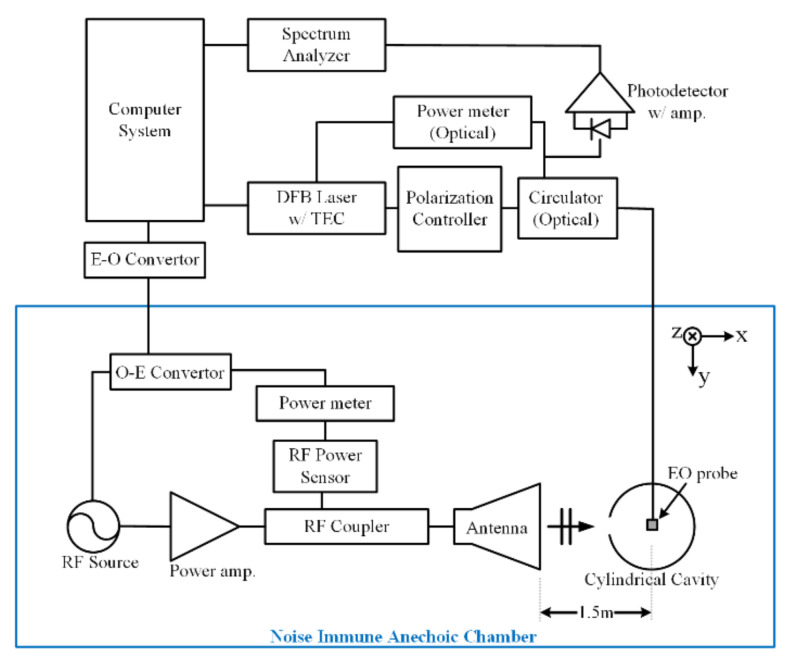
Experimental block diagram of the EO measurement system with a noise immune anechoic chamber.

**Figure 6 sensors-23-03278-f006:**
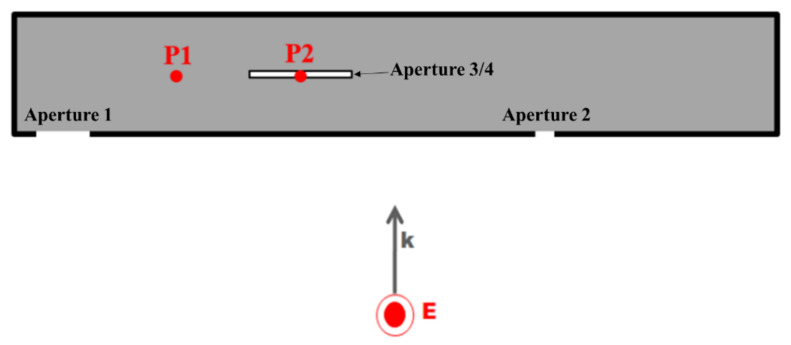
Measurement and simulation positions P1 and P2.

**Figure 7 sensors-23-03278-f007:**
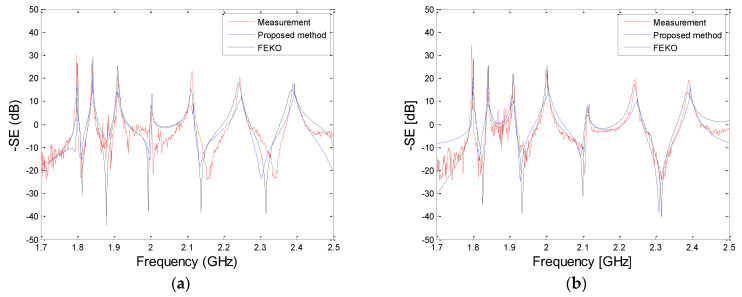
Comparison of the E−field Ey between the measurement and the proposed method: (**a**) P1; and (**b**) P2.

**Table 1 sensors-23-03278-t001:** Aperture sizes and positions on the cavity.

Aperture Index	Aperture Size	Center of Each Aperture Position (*ρ*, φ, z)
Aperture 1	46.8 mm × 15 mm	49.25 mm, 0, 607 mm
Aperture 2	17.72 mm × 17.72 mm	49.25 mm, 0, 207 mm
Aperture 3	88.75 mm × 5 mm	49.25 mm, π/2, 408.6 mm
Aperture 4	88.75 mm × 5 mm	49.25 mm, −π/2, 408.6 mm

**Table 2 sensors-23-03278-t002:** Comparison of the computing time between the proposed method and FEKO.

Analysis Method	Computing Time (s)
Proposed method	62
FEKO (MoM based)	6240

## Data Availability

This study did not report any data.

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
