# Peer review of "Analysis of Intentional Electromagnetic Interference on GENEC Model Using Cylindrical Mode Matching"

_sensors, 2023, doi:10.3390/s23063278_

Round 1

Reviewer 1 Report

This paper proposes a new cylindrical mode matching (CMM) technique to analyze the generic missile (GENEC) model, a cylindrical PEC target. Using the CMM, we analyze the effect of the IEMI on a GENEC model with several apertures. The results were compared to those of the FEKO for verification, and showed good consistency. Overall, this paper is clear, well-written, and is suitable for sensors. I suggest publication, if the authors can make all the figures centerlized.

Author Response

Reviewer’s Comments :

This paper proposes a new cylindrical mode matching (CMM) technique to analyze the generic missile (GENEC) model, a cylindrical PEC target. Using the CMM, we analyze the effect of

the IEMI on a GENEC model with several apertures. The results were compared to those of the FEKO for verification, and showed good consistency. Overall, this paper is clear, well written, and is suitable for sensors.

I suggest publication, if the authors can make all the figures centerlized.

ANS:

All figures are centered as your comment.

Thank you for your kind comments that will be of great help to improve the quality of the paper.

Reviewer 2 Report

The generic missile (GENEC) model with several apertures is analyzed using a cylindrical mode matching (CMM) technique in the manuscript. For verification, the results were compared to experimental data and those of the FEKO; all exhibited good consistency. But, after reading this article, I have the following suggestions:

1. Special characters should be remarked when they first appear.

Line 22, Page 1, "PEC" should be remarked when it first occurs in the text. Line 24, Page 1, "FEKO" should be remarked when it first occurs in the text.

2. The pictures of the paper should be better presented. Readers can only get limited information. The details of the 4 apertures in Figure 2 are not clear. Some notes should be added in Figure 3 to help readers understand the experimental design of the paper. The clarity of the picture needs to be improved.

3. The latest research progress in related fields is not mentioned in the paper, which makes it impossible to determine the novelty of the model. In order to make your work more convincing, please add more than 5 citations from other scholars or research group in recently years.

4. A large number of formulas are quoted in this paper, which is difficult to understand. It is suggested to put some unimportant formulas in the supporting files, such as the derivation process of some formulas. This can help readers read the article more easily.

Author Response

Reviewer’s Comments:

The generic missile (GENEC) model with several apertures is analyzed using a cylindrical mode matching (CMM) technique in the manuscript. For verification, the results were compared to experimental data and those of the FEKO; all exhibited good consistency. But, after reading this article, I have the following suggestions:

  1. Special characters should be remarked when they first appear. Line 22, Page 1, "PEC" should be remarked when it first occurs in the text. Line 24, Page 1, "FEKO" should be remarked when it first occurs in the text.

  1. The pictures of the paper should be better presented. Readers can only get limited information. The details of the 4 apertures in Figure 2 are not clear. Some notes should be added in Figure 3 to help readers understand the experimental design of the paper. The clarity of the picture needs to be improved.

  1. The latest research progress in related fields is not mentioned in the paper, which makes it impossible to determine the novelty of the model. In order to make your work more convincing, please add more than 5 citations from other scholars or research group in recently years.

  1. A large number of formulas are quoted in this paper, which is difficult to understand. It is suggested to put some unimportant formulas in the supporting files, such as the derivation process of some formulas. This can help readers read the article more easily.

ANS:

  1.  

- The apertures in Figure 2 have been modified for clarity.

- Figure 3 is a picture showing the real measurement environment, and more comments have been added to enhance understanding the figure.

  1. We added 5 papers as citation references.

  1. First of all, we are sorry for the confusion in formula development. However, since the formula development is the core of this article, we think that the reduction of the number of equations may cause problems in theoretical understanding.

Thank you for your kind comments that will be of great help to improve the quality of the paper.

Reviewer 3 Report

The manuscript does not contain any significantly new information on the topic and therefore cannot be published as a communication. The authors are encouraged to compare the calculations with other models and perform measurements and simulations for a few more cavity dimensions and positions from the center of the model to understand the model limitations.

Author Response

Reviewer’s Comments:

The manuscript presents a cylindrical mode matching (CMM) technique to analyze the generic missile (GENEC) model with several apertures. The results were compared with experimental data and those of the FEKO for verification, and all showed good consistency.

The manuscript is well organized and clearly presented with detailed mathematical modeling and experimental data so it can be considered for publication, Yet I would recommend a final check for English minor mistakes

ANS:

We modified the article to make it easier to understand by correcting redundant or unnatural sentences.

Thank you for your kind comments that will be of great help to improve the quality of the paper.

Reviewer 4 Report

The manuscript presents a cylindrical mode matching (CMM) technique to analyze the generic missile (GENEC) model with several apertures. The results were compared with experimental data and those of the FEKO for verification, and all showed good consistency.

The manuscript is well organized and clearly presented with detailed mathematical modeling and experimental data so it can be considered for publication, Yet I would recommend a final check for English minor mistakes.

Author Response

Reviewer’s Comments:

This manuscript provides a cylindrical mode matching (CMM) to calculate the GENEC model). The model is straight forward and comparison with the previous simulation and experimental results is reasonable. However, there are a few comments after reviewing this manuscript:

  1. This manuscript missed the units for all variables which should be included.

  1. It is suggested to mention more about the findings of this study at the end of abstract

  1. I strongly recommend the authors to add one paragraph discussing the difference between their work and the previously performed studies in literature. In other words, what is the novelty of this work? I offer the authors to revise the abstract and introduction in order to incorporate the novelty of their work. This change motivates the readers of sensors to study this work with interest.

  1. There are too many professional abbreviations, which makes the whole article difficult for readers to understand, and the references are too old.

  1. The details in Figure 2 do not show how the holes of the cylinder are designed clearly. It is recommended to mark them with different colors. In the experiment in Figure 3, the extra electromagnetic wave is emitted by what direction, you also can mark in the figure, Figure 6 is very confusing to readers, because there are not any legend.

ANS:

  1. Units for electric and magnetic field vectors have been added as your comment
  2. We added more mention to the abstract as your comment.
  3. We added more references related to our research and explained the difference from these cited references.
  4. We added the recent references. When using abbreviations for the first time, we wrote their full words together. (PEC, EMT, BLT, …)
  5. The apertures in Figure 2 have been modified to better represent 4 apertures.

Figure 3 is a picture showing the real measurement environment, and more comments have been added to enhance understanding the figure.

In Figure 6, we have added some legends for 4 apertures.

Thank you for your kind comments that will be of great help to improve the quality of the paper.

Reviewer 5 Report

This manuscript provides a cylindrical mode matching (CMM) to calculate the GENEC model).  The model is straight forward and comparison with the previous simulation and experimental results is reasonable. However, there are a few comments after reviewing this manuscript:

1. This manuscript missed the units for all variables which should be included.

2. It is suggested to mention more about the findings of this study at the end of abstract.

3. I strongly recommend the authors to add one paragraph discussing the difference between their work and the previously performed studies in literature. In other words, what is the novelty of this work? I offer the authors to revise the abstract and introduction in order to incorporate the novelty of their work. This change motivates the readers of sensors to study this work with interest.

4. There are too many professional abbreviations, which makes the whole article difficult for readers to understand, and the references are too old.

5. The details in Figure 2 do not show how the holes of the cylinder are designed clearly. It is recommended to mark them with different colors. In the experiment in Figure 3, the extra electromagnetic wave is emitted by what direction, you also can mark in the figure, Figure 6 is very confusing to readers, because there are not any legend. 

Author Response

(The authors gave the same response as above.)

Round 2

Reviewer 3 Report

The manuscript could be published in its current form.